# Calibrating a Comprehensive Immune Age Metric to Analyze the Cross Sectional Age-Related Decline in Cardiorespiratory Fitness

**DOI:** 10.3390/biology11111576

**Published:** 2022-10-27

**Authors:** Peter Bröde, Maren Claus, Patrick D. Gajewski, Stephan Getzmann, Klaus Golka, Jan G. Hengstler, Edmund Wascher, Carsten Watzl

**Affiliations:** Leibniz Research Centre for Working Environment and Human Factors at TU Dortmund (IfADo), Ardeystrasse 67, D-44139 Dortmund, Germany

**Keywords:** aging, immunosenescence, physical fitness, physical activity, obesity, sex

## Abstract

**Simple Summary:**

Cardiorespiratory fitness as a crucial prerequisite for sustained work ability declines with aging, as does the functionality of the immune system, the latter process termed immunosenescence or immune age. We approximated a comprehensive immunosenescence biomarker by just a few flow-cytometry-based parameters using blood samples. Applied to measurements with 597 participants from the Dortmund Vital Study, we could show that immune age, but not chronological age, together with obesity and physical inactivity, independently from sex, were significant predictors for the probability of low cardiorespiratory fitness.

**Abstract:**

Cardiorespiratory fitness (CRF) is essential for sustained work ability in good health, but declines with aging, as does the functionality of the immune system, the latter process commonly referred to as immunosenescence. This study aimed to compare the capacity of immunosenescence biomarkers with chronological age for predicting low CRF in a cross-sectional sample recruited from the regional working population. CRF was determined by submaximal bicycle ergometer testing in a cross-sectional sample of 597 volunteers aged 20–70 years from the ’Dortmund Vital Study’ (DVS, ClinicalTrials.gov Identifier: NCT05155397). Low CRF was scored if the ergometer test was not completed due to medical reasons or if the power output projected to a heart rate of 130 bpm divided by body mass was below sex-specific reference values of 1.25 W/kg for females and 1.5 W/kg for males, respectively. In addition to established biomarkers of immunosenescence, we calibrated a comprehensive metric of immune age to our data and compared its predictive capacity for low CRF to chronological age, while adjusting our analysis for the influence of sex, obesity, and the level of regular physical activity, by applying univariate and multiple logistic regression. While obesity, low physical activity, chronological and immune age were all associated with increased probability for low CRF in univariate analyses, multiple logistic regression revealed that obesity and physical activity together with immune age, but not chronological age, were statistically significant predictors of low CRF outcome. Sex was non-significant due to the applied sex-specific reference values. These results demonstrate that biological age assessed by our immunological metric can outperform chronological age as a predictor for CRF and indicate a potential role for immunosenescence in explaining the inter-individual variability of the age-related decline in cardiorespiratory fitness.

## 1. Introduction

Cardiorespiratory fitness (CRF) helps prevent cardiovascular disease and premature mortality [1,2,3,4], and is crucial for sustained work ability in good health [5,6,7,8,9] in both physically [10,11] and cognitively demanding occupations [12]. CRF exhibits an interdependent relationship with physical activity and the immune system [8,13,14,15], where physical activity and exercise help to improve CRF, inhibiting inflammatory responses [16], thus strengthening and maintaining the functioning of the immune system during aging [17,18]. Aging individuals in western societies frequently exhibit a sedentary lifestyle, characterized by physical inactivity, especially after retirement from work. This may lead to obesity and a low level of CRF, elevating health risks [19]. Concomitantly, the age-related CRF decline [20] profoundly varies in different groups defined by individual characteristics such as sex, body composition, obesity, and health status [21,22].

Hence, the assessment of CRF forms an integral part of the ongoing ’Dortmund Vital Study’ (DVS, ClinicalTrials.gov Identifier: NCT05155397), a combined cross-sectional and longitudinal interdisciplinary study using as study sample a cohort of 600 individuals recruited from the regional working population. The DVS aims at investigating the complex interplay of aging, working conditions, genetics, stress, metabolism, cardiovascular system, immune system, and physical and mental performance over the course of the working life of healthy adults. While companion papers describing the detailed study protocol [23] and broad analyses of sociodemographic, biological, and environmental influences on lifespan work ability [24] are available elsewhere, this report will focus on factors contributing to the inter-individual variability in the age-related decline of CRF.

Recently, metrics of ‘biological age’ have gained attention or even outperformed chronological age [25,26] as predictors for age-related mortality, health, and disease [27,28,29], the success of vaccination in the elderly population [30], or declining brain function [31]. Likewise, the concept of ’immune age’ or ‘immunosenescence’ aims at quantifying, preferably by a one-dimensional marker, the decay in functions of the immune system with individually varying progression in the elderly [32,33], which does not necessarily parallel chronological age [34,35].

Several immunosenescence biomarkers have been proposed, such as memory/naïve sub-populations of CD4^+^ and CD8^+^ T-cells, the CD4/CD8-ratio, or the number of CD28^–^ T-cells with application in predicting age-related morbidity and mortality [36,37,38,39,40,41,42,43,44]. More recently, these biomarkers were complemented by attempts to define composite scores aiming at a comprehensive assessment of the aging immune system [35,45,46].

The immune age metric IMM-AGE [35] was recently built from longitudinally following the immune status of 135 healthy volunteers for up to nine years employing multi-omics techniques to a high dimensional set of parameters, comprising blood cell phenotypes, functional tests with stimulated cells, and gene-expression analyses. Thus, IMM-AGE has been widely recognized as a cutting-edge biomarker comprehensively covering the processes related to immunosenescence [25,34,47,48,49,50,51,52,53]. However, applications to epidemiological and clinical settings are scarce, because a one-to-one implementation of the published procedure would not only demand advanced and extensive analytics, but also require to mimic the longitudinal sampling scheme, as illustrated by corresponding complaints in a recent study concerning SARS-CoV-2 [54]. Even in the original study [35], IMM-AGE had to be approximated by a compatible set of gene expression parameters for demonstrating its applicability to cardiovascular health data from the Framingham cohort.

In order to apply such advanced immunosenescence biomarkers for investigating the relation of cardiorespiratory fitness with the aging immune system in the DVS, we present a simplified method to determine a novel metric for immune age using a limited set of flow cytometry-based immune parameters. To do so, we scrutinized the immune data published with the original IMM-AGE study [35] for compatibility with the immune parameters measured in the DVS. For calibrating the comprehensive IMM-AGE metric to our data, we used this set of compatible variables to determine an approximate score for application as predictor within the DVS. In order to distinguish the novel marker from the original metric, we termed the approximation IMMAX (IMMune Age indeX).

Aiming on factors contributing to the inter-individual variability in the age-related decline of CRF, with a special focus on the role of immunosenescence, the present analysis compares the capacity of IMMAX with chronological age for predicting low CRF in a cross-sectional sample from the DVS when adjusting for sex, obesity and regular physical activity.

## 2. Materials and Methods

### 2.1. DVS Sample

The DVS (ClinicalTrials.gov Identifier: NCT05155397) is conducted with approval from the local Ethics Committee of the IfADo [23]. It is designed as a combined cross-sectional and longitudinal study comprising a baseline and up to three follow-up examinations, separated by five-year intervals with a projected completion of data collection by end of 2035. Here, we analyzed cross-sectional data of the baseline examinations performed between 2016 and 2021, consisting of observations from 368 females and 229 males aged 20–70 years.

As detailed in the published study protocol [23], the sample was drawn from the general healthy regional working population, and deemed representative concerning age, genetics, and occupation, whereas females were slightly overrepresented [23]. By structured pre-study telephone interviews, we collected information on health status. Applying a wide scope for ‘healthy’, we also included individuals who were smokers, normal alcohol users (no alcohol use disorder), overweight, or persons with non-severe disease symptoms, allowing for medications with, e.g., anticoagulants, hormones, or antihypertensive and cholesterol-lowering drugs. On the other hand, we excluded persons with severe neurological, cardiovascular, or oncological diseases and psychiatric disorders (cf. [23] for a comprehensive list) from participation.

From the measured body mass (in kg) and height (in m), body composition (obesity) was assessed by the body-mass index (BMI = mass/height^2^) and scored as normal for BMI < 25 kg/m^2^, as overweight for 25 kg/m^2^ ≤ BMI ≤ 30 kg/m^2^ and as obese for BMI > 30 kg/m^2^, respectively [55].

Regular physical activity was assessed by the Lüdenscheid Physical Activity Questionnaire [56], an instrument with well-proven utility in recent studies [57,58,59]. This questionnaire consists of 13 items about physical activity during work and leisure time, which are summarized into a four-level score with respect to preventing health risks associated with inactivity (1: (too) low, 2: still acceptable, 3: satisfactory, 4: high).

Table 1 presents the DVS sample distribution of individual characteristics together with covariates and outcomes as described below.

### 2.2. Cardiorespiratory Fitness Assessment

CRF was operationalized by the result of the physical working capacity test PWC130 [60], a submaximal incremental testing procedure on a bicycle ergometer. Following standard recommendations [60,61,62] and adhering to the requests of the institute’s Ethics Committee [23], all cycle ergometer tests were performed under medical supervision. The participants cycled with a cadence of 60 rpm (revolutions per minute) starting with 25 W required power output, which was increased every 2 min by 25 W, until the participants’ heart rate as recorded by electrocardiography (ECG) exceeded 130 bpm. The PWC130 outcome was defined as the power output projected to a heart rate of 130 bpm divided by body mass (in W/kg).

PWC130 outcome was missing for more than 15% of the observations (Table 1), with only less than one third being attributable to technical issues with the equipment or to the non-availability of medical supervision. The majority of missing observations was associated with non-performing or stopping the test prematurely, i.e., before reaching the projected heart rate of 130 bpm due to medical reasons, like abnormal ECG recordings, hemodynamic changes, being exhausted, or medical contraindications [60,62]. As these missing observations were indicative for low CRF and thus considered non-ignorable, CRF was quantified by the dichotomization of the PWC130 outcome. Low CRF was scored if the participant could not complete the test due to medical reasons, with such events recorded in the study log, or if the PWC130 outcome was below a sex-specific reference value of 1.25 W/kg for females and 1.5 W/kg for males, respectively [63]; otherwise, high CRF was scored. This approach reduced the number of missing observations considerably (Table 1).

### 2.3. Immune Parameters

Peripheral venous blood (80 mL) was collected from DVS participants in heparinized monovettes (Sarstedt, Nümbrecht, Germany) and a set of relative blood cell frequencies was determined by flow cytometry [64]. Briefly, we isolated peripheral blood mononuclear cells (PBMC) by Ficoll density gradient centrifugation (PAN-Biotech, Aidenbach, Germany), and cells were stored at −170 °C for up to 6 months until analysis. Four antibody panels were built to gain information on the general lymphocyte and monocyte subpopulations and to analyze the lymphocytes for NK/T cell ratio, CD4/CD8 T cell ratio, memory/naïve sub-populations of CD4^+^ and CD8^+^ T cells, and CD28^−^ T cells, which are all related to aging and senescence. All antibodies were individually titrated to determine the optimal dilution. All antibodies and dilutions are listed in the Appendix A by Appendix A. Gating strategy is presented in Appendix A. We stained PBMC immediately after thawing and kept them on ice during the entire procedure. For each panel, we stained 0.2 × 10^6^ cells with the indicated antibody cocktails for 20 min in the dark at 4 °C and afterwards washed them with FACS buffer (PBS/2% FCS). Cells were resuspended in FACS buffer and kept on ice until analysis at the same day on a BD LSRFortessa. Data were analyzed using the FlowJo software (FlowJo LLC, Ashland, OR, USA).

Accounting for the compositional structure of related relative cell frequencies, e.g., memory and naïve CD8 T-cells, which inherently exhibit a negative correlation because their sum is limited by 100%, we transformed such pairs to their log-ratio, while single percentage cell frequencies (*%p*) were transformed to their *logit*(%*p*) = *log*(%*p*/(100% − %*p*)) [65].

### 2.4. Approximating IMM-AGE by IMMAX

We obtained 434 values of the IMM-AGE metric normalized to the range between zero and one from the published Appendix A [35], which were merged with chronological age and raw cellular relative frequencies of 65 variables listed in Appendix A, whereas the data did not contain any information about the sex of the participants.

We screened our set of immune parameters from the DVS for compatibility with the original study and compared the distribution of these candidate variables between the two studies. We adjusted our analyses for chronological age by linear regression after transforming the candidate variables to their *log-ratio* or *logit* as before, and subsequently evaluated the correlation coefficients and regression lines. The set of comparable immune parameters, as identified according to this procedure, were then used as predictors of IMM-AGE in a principal component regression model fitted by the package *pls* [66] using the R version 4.2.1 (R Core Team, Vienna, Austria) [67]. The resulting predicted scores were termed IMMAX (IMMune Age indeX), a one-dimensional metric to describe the immune age. Before fitting the regression model, we transformed the dependent variable to its *logit*, thus ensuring that IMMAX stayed between 0 and 1 just as IMM-AGE [68]. The Appendix A includes a corresponding R script as Appendix A.

### 2.5. Statistical Analysis of Cardiorespiratory Fitness

In the DVS sample we analyzed the capacity of sex, obesity, physical activity, age, and various immunosenescence biomarkers (IMMAX plus its predictors from the principal component regression model), for predicting the probability of low CRF by fitting univariate and multiple logistic regression models using the R function *glm* [69]. Only the 547 records with complete observations for the covariates and CRF, which comprised 199 low CRF events, and were included in the analyses. The estimated coefficients were expressed as odds ratios (OR) with 95%-confidence intervals (CI). For comparison purposes, the continuous predictors (age and immunosenescence biomarkers) were z-standardized to zero mean and unit variance prior to analyses; with in such a way standardized ORs representing the effect of 1 SD increase in the predictor. Model fit was assessed by Pearson’s correlation coefficient (R), the root mean squared prediction error (RMSE) and Akaike’s information criterion AIC [70].

## 3. Results

### 3.1. IMM-AGE Approximation

To simplify the determination of a comprehensive immune age metric, we first scrutinized the list of raw cell frequency variables of the original IMM-AGE study [35] for compatibility with our immune parameters from the DVS and identified 16 candidates for further inspection (Appendix A), with several variables showing considerable correlation with IMM-AGE in the original data (Appendix A). Comparing the distribution of these variables between the two studies revealed significant differences for many immune parameters (Appendix A). However, the chronological age also differed significantly between the two studies. The IMM-AGE study included two distinct age groups, young adults (20–36 years) and older adults (63–97 years), whereas participants in the DVS ranged from 20–70 years (Figure 1A). Hence, after excluding variables with a high proportion of missing values (Appendix A), we adjusted our analyses for chronological age by applying linear regression models. The comparison of age-dependent regression lines and correlation coefficients between the two studies for the remaining 12 candidate predictors (Appendix A) suggested a reduced set of compatible peripheral blood mononuclear cell sub-populations (NK-cells, T cells, total and memory/naïve sub-populations of CD4 and CD8 T-cells, CD8 CD28^–^ T-cells). From these, we calculated five immunosenescence biomarkers on a log-ratio or logit scale (Figure 1B), which were then used as predictors in the principal component regression model. The estimated coefficients (Appendix A) allowed for calculating a simplified score for the immunological age, which we termed IMMAX (Immune Age Applied). IMMAX was determined not only for the original sample [35], but also for the DVS [23,24].

For the sample from the original study [35], our IMMAX score approximated the primary IMM-AGE values with acceptable accuracy, as indicated by high Pearson’s correlation R = 0.84 and moderate typical prediction error RMSE = 10% (Figure 1C). The regression lines with chronological age agreed well for IMM-AGE and IMMAX from the original study and were in line with the regression function of the approximation for the DVS (IMMAX.DVS), which nicely filled the gap in the bimodal age distribution (Figure 1D). Figure 1E presents the regression lines with age separately for females and males in the DVS, which were parallel, as indicated by a non-significant interaction term (*p_age*sex_* = 0.97), increased by 0.43% per year and were shifted downwards for females by 5.1%, corresponding to a horizontal shift of approximately 12 years (5.1%/0.43%/year). Similar results demonstrating that females are immunologically younger compared to males with the same chronological age had been reported for the Framingham cohort in the original study [35], thus pointing to the validity of our approximation.

### 3.2. Associations with Cardiorespiratory Fitness

#### 3.2.1. Univariate Analyses

In the DVS sample, obesity status, a low level of regular physical activity, chronological age, and the immune age metric IMMAX, as well as the ratio of memory to naïve CD8 cells and CD8 CD28^–^ cells, correlated positively with low CRF, as illustrated by Figure 2, and assessed in the univariate analyses corrected for multiple testing presented by Table 2. Notably, the ratios of NK- to T-cells, of CD4 to CD8 cells, and of memory to naïve CD4 cells, showed no associations with low CRF. Although males reached higher relative power output compared to females in the PWC130 (Table 1), sex was not associated with low CRF, which could be expected since the dichotomization was based on sex-specific reference values [63].

#### 3.2.2. Multivariate Analyses

Figure 3A presents the outcome of the multiple logistic regression analyses focusing on the contrast between chronological age with the immune age metrics in terms of the standardized odds ratios and the AIC for model comparison, while the detailed results comprising all covariates are shown by Appendix A. They revealed that obesity and level of regular physical activity together with immune age, but not chronological age, were statistically significant predictors of low CRF. Remarkably, the ratio of memory to naïve CD8 cells was competitive to the comprehensive metric IMMAX concerning predictive capacity, as indicated by a slightly higher standardized OR accompanied by a lowered AIC (Figure 3A), which, however, showed an absolute difference below 2 (Appendix A) indicating a comparable fit for the two models [70].

Figure 3B presents the corresponding outcomes from fitting a slightly modified series of models separately including age and the six immune age metrics as predictors, but omitting the non-significant factor sex from the analyses. Detailed information for all covariates is provided in Appendix A and confirmed the previous results. In particular, chronological age became a non-significant predictor when adjusting for covariates, while the immune age metric IMMAX as well as the ratio of memory to naïve CD8 cells and CD8 CD28^–^ cells persisted as significant predictors for low CRF, while the metrics involving CD4 or NK cells were non-significant, which agreed with the univariate analysis (Figure 2). Note that omitting sex and age as non-significant predictors in the models involving immunosenescence biomarkers lowered the corresponding AIC values, compared to the analyses presented by Figure 3A, thus improved the model fit.

## 4. Discussion

### 4.1. Calibrated Immune Age Metric IMMAX

For our study, we adopted the advanced immune age metric IMM-AGE [35] through approximation by a set of peripheral blood mononuclear cell frequencies from flow cytometry to determine the IMMAX metric, which showed a reasonable prediction error of 10%. This approximation exhibited a relationship with chronological age for the DVS data that was in line with corresponding relations from the original study. Additionally, it revealed a 5% reduction in immune age for females compared to males of identical chronological age, which corresponded to a 12-years shift, thus confirming earlier findings of lower immune age for females [35,43]. These outcomes point to the validity of our approximation, which was then applied to predict low CRF in comparison to established immunosenescence biomarkers and to chronological age.

### 4.2. Cardiorespiratory Fitness and Immune Age

Submaximal cycle ergometer tests, such as the PWC130, have been widely applied to CRF assessment in observational research [71] because their results highly correlated with the output of procedures requiring maximal exertion [72], but they are less strenuous and more likely to be completed, especially in an elderly study population. Furthermore, the submaximal test output can be considered as a physical performance measure [72], and was recently shown to detect changes in CRF in longitudinal settings with reasonable precision [73]. Thus, the submaximal PWC130 was the method of choice for assessing CRF in the combined cross-sectional and longitudinal DVS.

The mean PWC130 outcomes in our study exceeded sex-specific reference values applied in sports medicine [63] and approximately corresponded to the 75th percentiles reported recently for a German cohort aged 45 to 64 years [71]. However, this does not necessarily indicate above-average physical working capacity, because our sample did also include younger persons, and in addition showed a considerable number of tests not completed due to medical reasons or contraindications. Low CRF was therefore assessed by dichotomization in order to avoid the bias potentially introduced by ignoring these informative missing observations.

While sex showed no significant effect due to the applied sex-specific reference values [63], our results are confirmative concerning the well-established detrimental influence of obesity and low level of regular physical activity on CRF [21,22]. As body mass might be considered a (often meagre) proxy to muscle mass, one might argue about improving the accuracy of the study results by using information on body fat content or fat-free mass instead of BMI and body mass for categorizing obesity and standardizing PWC130 output, respectively. However, it should be noted that the quantities applied in this study constitute established markers of obesity and CRF, with wide application in observational studies and existing reference values [19,20,21,55,63,71]. In addition, as the effects of obesity on CRF in this study were quite stable in the various univariate and multivariate analyses, this may indicate that, with the reservations mentioned above, using body mass for standardization of the PWC130 output and applying BMI for categorizing obesity allowed for an adequate consideration of obesity as covariate in our analyses focusing on the role of immunosenescence in explaining the decline in CRF with age.

The age-related increase in the probability for low CRF, which was observed in univariate analyses, vanished when adjusting for the covariates, indicating that a decline of CRF with age observed at the population level might be linked to a mutual interplay between physical inactivity, often associated with a sedentary lifestyle, with obesity and the immune system [8,13,14,15,22].

On the other hand, the statistical significance of the approximated advanced immune age metric IMMAX persisted in multivariate analyses. In particular, replacing chronological age by immune age as a predictor for low CRF lowered the AIC, i.e., increased the predictive capacity of the models (Figure 3). These findings are in line with previous reports on markers of ‘biological age’ superseding chronological age as a predictor for morbidity and mortality in aging populations [25,28,35]. Remarkably, concerning the predictive capacity for low CRF, the ratio of memory to naïve CD8 cells performed on a level competitive to the advanced metric IMMAX. This confirms the role of the age-related decrease of peripheral naïve cells accompanied by the accumulation of memory T-cells, especially in the CD8 subpopulation, as established markers of immunosenescence [38,40,74,75]. This is also supported by the high correlation between naïve CD8 cells and the IMM-AGE metric in the original study (Appendix A). As the downregulation of costimulatory molecule CD28 with age leading to progressive expansion of CD28^−^ cells has been considered as a ‘compensation’ for the reduction of naïve CD8 cells [40] and a hallmark of senescence [37], this may explain the somewhat lower, but significant associations found for CD8 CD28^–^ cells. No significant associations with cardiorespiratory fitness occurred for markers involving CD4 or NK cell subpopulations, although associations with CD4 cells had been reported before [15], as well as acute exercise effects on NK cell frequency and function [18,76].

Though we found that memory/naïve CD8 T cell ratio and the new IMMAX metric have comparable predictive capacity, this does not necessarily imply a direct link between the functions of CD8 T cells and cardiorespiratory fitness. More likely, the same factors influencing memory T cells (infections and other immune challenges) will have a negative impact, e.g., on general health status and age-related morbidity [44], which in turn may influence the level of physical activity and cardiorespiratory fitness in an interdependent manner [13,14,15]. While the likelihood of such immune challenges increases with chronological age, our immune age metric is a more direct measure of these events, possibly explaining why immune age is a better predictor than chronological age for cardiorespiratory fitness in our study.

### 4.3. Outlook

The ongoing longitudinal examinations within the DVS cohort will allow for the verification of our cross-sectional results considering that the intertwined relationships between cardiorespiratory fitness, regular physical activity, and the immune system, will advocate for longitudinal studies [8,13,27]. In addition, with this study providing a simple assay system to determine IMMAX as a comprehensive metric for immunological health, this opens up the possibility to assess the immune age in future studies and enables studying correlations of immune age with other physiological and psychological outcomes [23,24].

## 5. Conclusions

In conclusion, our results indicate a potential role for the immune age in explaining the inter-individual variability of the age-related decline in cardiorespiratory fitness. This may have implications for work ability and prevention concerns in occupational health and safety practice, e.g., for CRF assessment in physical employment standards [6]. Here, our approach might be instructive on how to approximate or even replace advanced immunosenescence biomarkers by less expensive methods involving cell subset frequencies, e.g., of naïve and memory CD8 cells.

## Figures and Tables

**Figure 1 biology-11-01576-f001:**
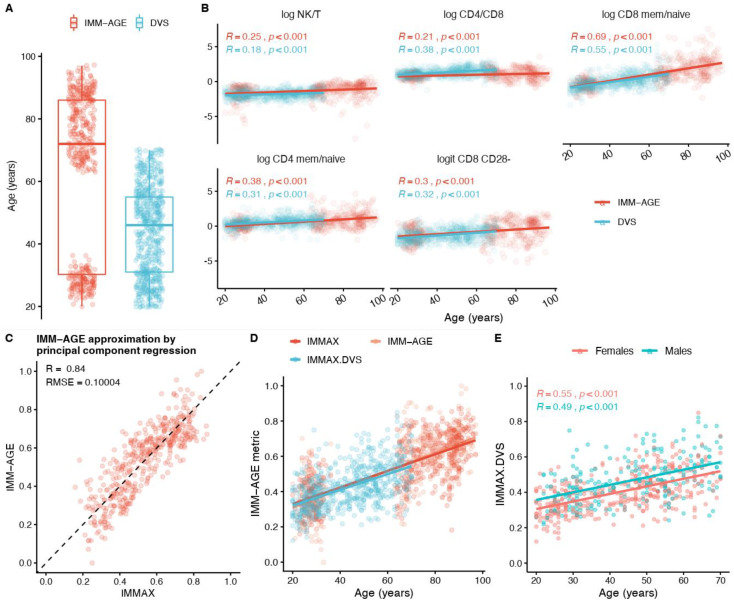
Calibration of the IMM-AGE metric [35] to cell-frequency data from the Dortmund Vital Study DVS [23]. (**A**) Box plots of the age distributions in the IMM-AGE sample and the DVS. (**B**) Compatibility between IMM-AGE and DVS for five biomarkers of immune age (NK- to T-cell ratio, CD4:CD8 ratio, memory-to-naive ratios for CD8 and CD4 T-cells, CD28- CD8 cells) in relation to chronological age assessed by linear regression and Pearson correlation coefficients (R). The analyses used the logarithms of ratios and the logits of percentages, respectively. (**C**) Goodness-of-fit in comparison to dashed line of identity assessed by Pearson correlation coefficient and root-mean-squared error (RMSE) of the approximation to the IMM-AGE metric in the original data [35] calculated by principal component regression (IMMAX) with the five biomarkers from (B) as predictors. (**D**) Age-depending linear regression lines for the IMM-AGE metric and its approximation (IMMAX) in the original data from (C) compared to the approximations calculated for the DVS data (IMMAX.DVS). (**E**) Linear regression and correlation with age of the approximated IMM-AGE metric in the DVS (IMMAX.DVS) for females and males.

**Figure 2 biology-11-01576-f002:**
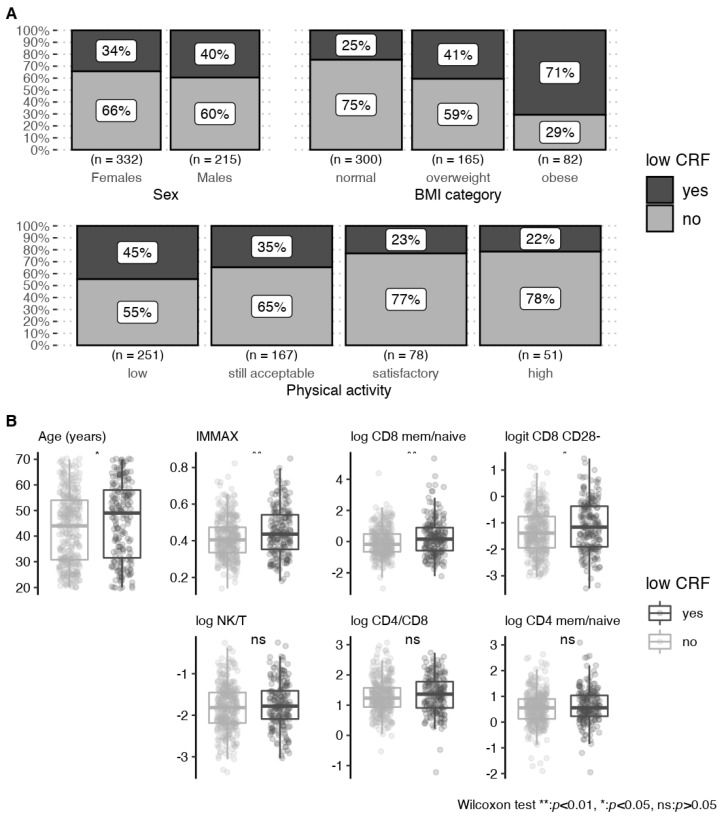
Bivariate associations of low cardiorespiratory fitness (CRF) with (**A**) categorical and (**B**) continuous predictors for the subsample of 547 complete observations from the DVS with 199 low CRF events.

**Figure 3 biology-11-01576-f003:**
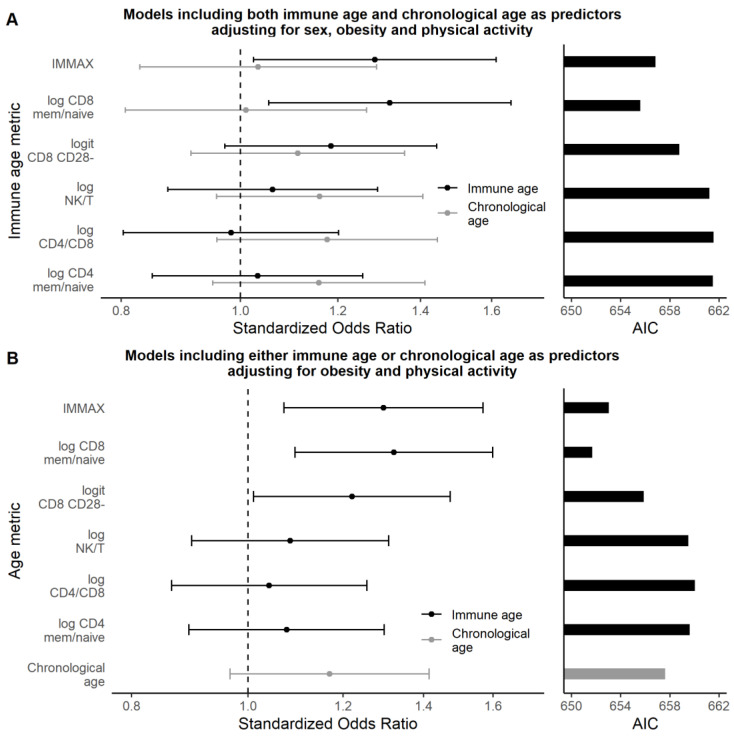
Multiple logistic regression results comparing chronological age with different immune age metrics as predictors of low CRF by standardized odds ratios with 95%-CI (left panels, with vertical dashed reference lines indicating null effect) and by Akaike’s information criterion AIC with lower values indicating improved model fit (right panels), respectively. (**A**) Results using different immune age metrics as predictors in separate models in addition to chronological age, adjusting the analyses for sex, obesity, and physical activity. (**B**) Results using either chronological age or different immune age metrics as predictor in separate models, adjusting for obesity and physical activity, but excluding sex as covariate.

**Table 1 biology-11-01576-t001:** Sample characteristics stratified by sex, and summarized by mean (SD) for continuous data and by frequency (percentage) for categorical observations, respectively.

Characteristic	Overall *n* = 597	Females *n* = 368	Males *n* = 229
**Age (years)**	44 (14)	43 (14)	46 (14)
**Body height (m)**	1.73 (0.09)	1.68 (0.07)	1.82 (0.07)
**Body** **mass** **(kg)**	77 (17)	70 (15)	88 (16)
*#missing*	3	2	1
**BMI category**			
*normal*	323 (54%)	228 (62%)	95 (42%)
*overweight*	181 (30%)	90 (25%)	91 (40%)
*obese*	90 (15%)	48 (13%)	42 (18%)
*#missing*	3	2	1
**Physical activity score**			
*low*	268 (47%)	181 (51%)	87 (39%)
*still acceptable*	174 (30%)	101 (29%)	73 (33%)
*satisfactory*	79 (14%)	42 (12%)	37 (17%)
*high*	53 (9%)	29 (8%)	24 (11%)
*#missing*	23	15	8
**PWC130 (W/kg)**	1.61 (0.47)	1.52 (0.42)	1.77 (0.50)
*#missing*	92	59	33
**low CRF events**	211 (37%)	122 (35%)	89 (40%)
*#missing*	25	20	5
**Immunosenescence biomarker**			
*IMMAX*	0.43 (0.12)	0.40 (0.11)	0.47 (0.13)
*log NK/T*	−1.81 (0.55)	−1.91 (0.53)	−1.64 (0.54)
*log CD4/CD8*	1.31 (0.59)	1.32 (0.55)	1.31 (0.64)
*log CD8 mem/naive*	0.06 (1.03)	−0.11 (0.94)	0.35 (1.12)
*log CD4 mem/naive*	0.57 (0.68)	0.45 (0.64)	0.75 (0.69)
*logit CD8 CD28-*	−1.26 (0.88)	−1.36 (0.83)	−1.11 (0.94)

***BMI***: body-mass index; ***PWC130***: power output from the physical working capacity test on the bicycle ergometer at 130 bpm standardized for body mass; ***CRF***: cardiorespiratory fitness; ***#missing***: number of missing observations; ***IMMAX***: approximation to IMM-AGE metric [35] by principal component regression, termed IMMune Age indeX; ***NK***: %natural killer cells; ***T***: %T cells; ***CD4***: %CD4-positive T cells; ***CD8***: %CD8-positive T cells; ***mem***: %memory T cells; ***naïve***: %naïve T cells; ***CD8 CD28-***: %CD28-negative CD8-positive T cells; ***log***: natural logarithm; ***logit***: transformation of a percentage (%*p*) by *logit*(%*p*) = *log*(%*p*/(100% − %*p*)).

**Table 2 biology-11-01576-t002:** Bivariate associations assessed by odds ratios from univariate logistic regression models predicting the probability of low CRF by sex, obesity (BMI category), level of regular physical activity, and by chronological age and six immune age metrics. The continuous predictors had been z-standardized to zero mean and unit variance prior to analysis, which was performed for the subsample of 547 complete observations with 199 low CRF events.

Predictor	OR ^a^	95% CI ^a^	*p*-Value	q-Value ^b^
**Sex**			0.22	0.24
*Females* ^c^	—	—		
*Males*	1.25	0.88–1.78		
**BMI category**			**<0.001**	**<0.001**
*Normal* ^c^	—	—		
*overweight*	2.09	1.39–3.14		
*obese*	7.38	4.34–12.9		
**Physical activity**			**<0.001**	**0.001**
*Low* ^c^	—	—		
*still acceptable*	0.66	0.44–0.99		
*satisfactory*	0.37	0.20–0.66		
*high*	0.34	0.16–0.67		
**Age** (standardized)	1.25	1.05–1.50	**0.012**	**0.020**
**Immune age metric** (standardized)				
*IMMAX*	1.37	1.15–1.64	**<0.001**	**0.001**
*log CD8 mem/naive*	1.36	1.14–1.64	**<0.001**	**0.001**
*logit CD8 CD28-*	1.28	1.07–1.52	**0.006**	**0.013**
*log NK/T*	1.15	0.96–1.37	0.12	0.17
*log CD4 mem/naive*	1.13	0.95–1.35	0.17	0.21
*log CD4/CD8*	1.10	0.92–1.31	0.29	0.29

^a^ OR = Odds Ratio, CI = Confidence Interval for OR; ^b^ False discovery rate correction for multiple testing; ^c^ Reference category.

## Data Availability

The DVS data presented in this study are available on request from the corresponding author. In particular, the research data policy of the DVS is outlined in the Research Data Management section of the study protocol [23]. Raw IMM-AGE data are available as Appendix A from the original publication [35]. An R script for accessing the IMM-AGE data and calculating IMMAX is included as Code S1 in the Appendix A of this manuscript.

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
