# Peer review of "Calibrating a Comprehensive Immune Age Metric to Analyze the Cross Sectional Age-Related Decline in Cardiorespiratory Fitness"

_biology, 2022, doi:10.3390/biology11111576_

Round 1

Reviewer 1 Report

The authors completed a cross-sectional study to compare the capacity of immunosenescence biomarkers with chronological age for predicting low cardiorespiratory fitness (CRF). This is indeed a very interesting topic, while the paper is well-written with adequate descriptions of the methods, data analyses and conclusions. However, there is no clear statement in the introduction on what is the aim of the current study (it is reported only in the abstract), even though the reasons and the methods to be used are well explained. Why was this study conducted (reported)? In other words, what was the aim and why? I understand that this is part of a larger study, however, a clear direction on what the study is looking for is needed.

I have also a major concern about the distribution of the participants. The health status of the participants is not described and not considering in the data analysis and modelling, which is crucial, given that CRF and physical activity levels are highly dependent on health status. This causes a series of problems in the methods used and interpretation of the outcomes. For instance, in the CRF test medical supervision is reported. Was this required for all participants, even for the healthy ones? What was the reason of medical supervision in the CRF test? Was there a health pre-screening before the CRF test? Similarly, there are reported “missing” data, with a general statement as following: “i.e. before reaching the projected heart rate of 130 bpm due to medical reasons, like abnormal ECG recordings, hemodynamic changes, being exhausted or medical contraindications”. These reasons should be recorded and taking into account in the data analysis, otherwise specific characteristics of the participants may drive the final outcome to a specific direction. As a result, the final outcome seems not plausible anymore.

Author Response

We like to thank this reviewer for the positive feedback concerning the general value of our paper and appreciate the constructive comments concerning the structure of the introduction, especially considering the purpose and aims of our analyses. Please note that the general objective of the Dortmund Vital Study is introduced in an early paragraph of section 1 as follows:

“Hence, the assessment of CRF forms an integral part of the ongoing ’Dortmund Vital Study’ (DVS, ClinicalTrials.gov Identifier: NCT05155397), a combined cross-sectional and longitudinal interdisciplinary study using as study sample a cohort of 600 individuals recruited from the regional working population. The DVS aims at investigating the complex interplay of aging, working conditions, genetics, stress, metabolism, cardiovascular system, immune system, and physical and mental performance over the course of the working life of healthy adults. While companion papers describing the detailed study protocol [23] and broad analyses of sociodemographic, biological and environmental influences on lifespan work ability [24] are available elsewhere, this report will focus on factors contributing to the inter-individual variability in the age-related decline of CRF.”

In order to clarify the aims, purpose and approach of our analysis, we have added the corresponding information to the final paragraph of the introduction section, which now reads:

“Aiming on factors contributing to the inter-individual variability in the age-related decline of CRF with a special focus on the role of immunosenescence, the present analysis compares the capacity of IMMAX with chronological age for predicting low CRF in a cross-sectional sample from the DVS when adjusting for sex, obesity and regular physical activity.”

Concerning medical supervision of the cycle ergometer tests, we confirm that medical supervision was required for all participants of our study, as suggested by corresponding standard procedures (cf. ref. [60-62]), and as it was also requested by our local Ethics Committee, because a physically strenuous test might constitute a potential health hazard. This is now stated in the revised subsection 2.2, which reads:

“Following standard recommendations [60-62] and adhering to the requests of the institute’s Ethics Committee [23], all cycle ergometer tests were performed under medical supervision. The participants cycled with a cadence of 60 rpm (revolutions per minute) starting with 25 W required power output, which was increased every 2 minutes by 25 W, until the participants’ heart rate as recorded by electrocardiography (ECG) exceeded 130 bpm.”

Concerning the health status of the study participants, we acknowledge that, although we had published the study protocol including the major sample characteristics previously (cf. ref [23]), the health status of the study sample should be described in greater detail within this manuscript. Consequently, we added a corresponding paragraph to sub-section 2.1 of the methods section, stating that our sample represents ‘the healthy regional working population’ with a broad definition of ‘healthy’, based on the pre-study health screening interviews. Therefore, considering our participants as ‘healthy’ in this sense, we did not include a ‘health status’ variable in our further analyses. Nevertheless, with some participants showing health issues, e.g. abnormal ECG recordings, during the bicycle ergometer tests, such events were recorded and considered in the analysis as described in response to the next comment concerning data analysis below. The new paragraph in 2.1 reads:

“As detailed in the published study protocol [23], the sample was drawn from the general healthy regional working population, and deemed representative concerning age, genetics, and occupation, whereas females were slightly overrepresented [23]. By structured pre-study telephone interviews, we collected information on the health status. Applying a wide scope for 'healthy', we also included individuals who were smokers, normal alcohol users (no alcohol use disorder), overweight, or persons with non-severe disease symptoms, allowing for medications with, e.g. anticoagulants, hormones, or antihypertensive and cholesterol-lowering drugs. On the other hand, we excluded persons with severe neurological, cardiovascular, or oncological diseases and psychiatric disorders (cf. [23] for a comprehensive list) from participation.”

Thank you also for emphasizing the important detail on the treatment of (some) missing PWC130 values in the data analysis. Indeed, simply disregarding the performance tests, which were non-completed due to medical reasons, and only analyzing the completed tests would bias the results towards higher performance scores. Therefore, in addition to report mean PWC130 outcome in Table 1, we analyzed the PWC130 data after dichotomization into a ‘low CRF’ score (yes/no). This was possible, because those ‘reasons’ had been recorded during the trials, as now stated in subsection 2.2. This data pre-processing allowed for taking into account the informative dropouts (of non-completed trials) into the further analyses concerning factors influencing CRF, as described in subsection 2.5, and further discussed in section 4.2. The corresponding modified part of subsection 2.2 reads:

“PWC130 outcome was missing for more than 15% of the observations (Table 1), with only less than one third being attributable to technical issues with the equipment or to non-availability of medical supervision. The majority of missing observations was associated with non-performing or stopping the test prematurely, i.e. before reaching the projected heart rate of 130 bpm due to medical reasons, like abnormal ECG recordings, hemodynamic changes, being exhausted or medical contraindications [60,62]. As these missing observations were indicative for low CRF and thus considered non-ignorable, CRF was quantified by dichotomization of the PWC130 outcome. Low CRF was scored, if the participant could not complete the test due to medical reasons, with such events recorded in the study log, or if the PWC130 outcome was below a sex-specific reference value of 1.25 W/kg for females and 1.5 W/kg for males, respectively [63]; otherwise, high CRF was scored. This approach reduced the number of missing observations considerably (Table 1).”

Please also refer to the attachment and the tracked changes detailed in the revised manuscript.

Reviewer 2 Report

Using BMI exclusively to define obesity is a mistake. BMI does not differentiate between fat mass and musculoskeletal mass. Muscular people can be considered overweight.

Expressing cardiorespiratory fitness based on PWC139/Kg of total body mass is a way to penalize people with more muscle mass and less fat mass. Relativizing cardio-respiratory fitness by thinking that people with more body mass are more powerful is another mistake. The fat mass does not generate pedaling force, nor does it increase the watts. There are people with the same total body mass who have different fat and musculoskeletal masses.

By reclassifying the subjects according to their percentage of fat mass, or relative fat mass, other values may possibly be obtained in the relationships in which the pernicious effect of obesity was more evident.

Although the authors have done that part of the work based on BMI, I do not think it is necessary to change it, but they should include some comment on this in the discussion.

Author Response

We like to thank this reviewer for the valuable comments on the role of muscle mass and body fat. We agree with this reviewer, that body mass only represents a (often meagre) proxy to muscle mass, while the latter is certainly decisive for the power output during ergometer testing, with further potential differentiation between large (leg) and small (arm) muscle groups relevant for different modes of exercise, e.g. when comparing effects of cycling with arm cranking in laboratory studies. On the other hand, epidemiological or observational studies, as the DVS presented in this manuscript, often rely on the (easier obtainable) body mass for scaling the power output, with the advantage, that reference values are more commonly available for in a such way (i.e. using body mass) standardized response variable (cf. refs [20,21,63,71]). Future studies would need to establish such reference values for using fat-free or lean body mass as denominator for the PWC130/kg quantity.

Similar controversies prevail with the use of BMI for assessing obesity, which might, as this reviewer has pointed out, potentially misclassify muscular persons as overweight (though probably less often as obese). As such persons are usually highly trained (i.e. probably exhibiting high power output in the ergometer test), this may then lead to an underestimation of the pernicious effects of overweight on CRF due to using BMI for obesity classification. On the other hand, it is important to note that this is somewhat balanced by the standard approach of using body mass instead of fat-free mass in the PWC130/kg calculation, as discussed above, because dividing power output by body mass instead of fat-free mass gives more penalty to fat persons compared to lean, but muscular persons, so that there might be some overestimation of the pernicious effects of overweight on CRF due to using body mass in PWC130/kg calculation counteracting the underestimation due to BMI use for categorizing overweight.

In addition, as before, BMI constitutes an established marker of obesity with standardized reference values and wide international application in observational studies (cf. refs [19,21,55,71]). Furthermore, in this study, obesity had the role of a covariate for adjusting the analyses, which were focusing on the age and immunosenescence effects. As the effects of obesity on CRF in this study were quite stable in the various univariate and multivariate analyses, this may indicate that using body mass for standardization of the PWC130 output and applying BMI for categorizing obesity allowed for an adequate consideration of obesity as covariate in our analyses focusing on the role of immunosenescence in explaining the age-related decline in CRF.

While body fat was not determined in this study and, thus, corresponding data were not available for analysis, which in addition was more focused on the role of immunosenescence, we appreciate the option offered by this reviewer to consider this topic more deeply in the discussion, which we did by adding these considerations to the corresponding paragraph discussing the influence of obesity on CRF in subsection 4.2, which now reads as:

“While sex showed no significant effect due to the applied sex-specific reference values [63], our results are confirmative concerning the well-established detrimental influence of obesity and low level of regular physical activity on CRF [21,22]. As body mass might be considered a (often meagre) proxy to muscle mass, one might argue about improving the accuracy of the study results by using information on body fat content or fat-free mass instead of BMI and body mass for categorizing obesity and standardizing PWC130 output, respectively. However, it should be noted that the quantities applied in this study constitute established markers of obesity and CRF with wide application in observational studies and existing reference values [19-21,55,63,71]. In addition, as the effects of obesity on CRF in this study were quite stable in the various univariate and multivariate analyses, this may indicate that, with the reservations mentioned above, using body mass for standardization of the PWC130 output and applying BMI for categorizing obesity allowed for an adequate consideration of obesity as covariate in our analyses focusing on the role of immunosenescence in explaining the decline in CRF with age.”

Please also refer to the attachment and the tracked changes detailed in the revised manuscript.

Reviewer 3 Report

The authors presented a novel metric for immune age using a limited set of flow cytometry-based immune parameters. They scrutinized the immune data published with the original IMM-AGE study for compatibility with the immune parameters measured in the DVS. They used a set of compatible variables to determine an approximate score for 3 of 15 applications as a predictor within the DVS and calibrate the comprehensive IMM-AGE metric to their data. The approximation IMMAX (IMMune Age indeX) was defined to distinguish the novel marker from the original metric. Thus, the current study aimed to compare the capacity of IMMAX with chronological age for predicting low cardiorespiratory Fitness (CRF) in the Dortmund Vital Study (DVS) when adjusting for sex, obesity, and physical activity. 

This study has the potential content to spark interest from the readers and advances in the immunosenescence biomarkers and CRF field. However, I have major concerns to be addressed.  

Introduction. The background is well presented, but some sentences about the interplay between a sedentary lifestyle and aging should be provided. Indeed, the decline of CRF should be often reported in the literature in the older population who is, in its majority, sedentary. Thus, the reduced CRF is a combined effect of aging and sedentarism and should be not attributed to "natural" aging alone. 

Methods - Cardiorespiratory fitness assessment. Why do the authors support the inclusion of a submaximal test rather than a maximal test? Is the test validated before the study? How is its accuracy compared to the gold-standard test? The authors must provide more elements to defend the feasibility of this test to assess the CRF. 

Results. The authors make some mistakes when considering CRF and physical activity as the same variable. CRF has been considered as the performance during an exercise test, while physical activity is often related to the daily physical activity status - considering labor and leisure activities, for example. What variable is taken by the authors? Please make it clear for the readers. 

Discussion. As suggested in the introduction, physical inactivity and aging should be considered in a mutual direction. It means that, in the sedentary older population, the physiological declines in the CRF are erroneously attributed to aging per se but a combined effect of sedentarism and aging. Thus, it is very hard to determine how factor reduces CRF - sedentarism or aging.

Author Response

We are grateful that this reviewer acknowledges the value of our study and we appreciate the constructive remarks and suggestions made in order to improve the content of this paper. Please refer to our detailed responses below.

Following the hint for the introduction, we included the aspect of sedentary lifestyle into the corresponding introductory paragraph, which now reads:

“Aging individuals in western societies frequently exhibit a sedentary lifestyle, characterized by physical inactivity especially after retirement from work. This may lead to obesity and a low level of CRF, elevating health risks [19]. Concomittantly, the age-related CRF decline [20] profoundly varies in different groups defined by individual characteristics like sex, body composition, obesity, and health status [21,22].”

Concerning the methods, we agree that validity and reliability of the submaximal test procedure is an important issue. Submaximal testing has been widely applied in observational studies, because the procedure is less strenuous for the participants, especially in an elderly population, while at the same time the outcomes of submaximal and maximal tests were highly correlated in corresponding methodological studies. In order to address this relevant topic, we have added a corresponding paragraph at the beginning of section 4.2 in the discussion including two new references to a highly cited review and a recent study on successful application of submaximal cycle ergometer testing in a longitudinal setting with relevance to our DVS. The new paragraph reads as:

“Submaximal cycle ergometer tests like the PWC130 have been widely applied to CRF assessment in observational research [71] because their results highly correlated with the output of procedures requiring maximal exertion [72], but they are less strenuous and more likely to be completed, especially in an elderly study population. Furthermore, the submaximal test output can be considered as a physical performance measure [72], and was recently shown to detect changes in CRF in longitudinal settings with reasonable precision [73]. Thus, the submaximal PWC130 was the method of choice for assessing CRF in the combined cross-sectional and longitudinal DVS.”

Added references:

  1. Noonan, V.; Dean, E. Submaximal Exercise Testing: Clinical Application and Interpretation. Physical Therapy 2000, 80, 782-807, doi:10.1093/ptj/80.8.782.
  2. Björkman, F.; Ekblom, Ö.; Ekblom-Bak, E.; Bohman, T. The ability of a submaximal cycle ergometer test to detect longitudinal changes in VO2max. BMC Sports Science, Medicine and Rehabilitation 2021, 13, 156, doi:10.1186/s13102-021-00387-w.

Thank you also for bringing to our attention that, concerning the results, the distinction between CRF and physical activity status requires clarification. We agree with this reviewer that CRF (here operationalized by the outcome of a bicycle ergometer test) and regular (or daily) physical activity (here operationalized by a questionnaire) define two distinctive variables. As stated in subsection 2.1 of the methods section we evaluated the regular physical activity status of the participants using a questionnaire considering labor and leisure activities, which yielded categories of regular physical activity with respect to the prevention of cardiovascular health risks of (1:(too) low, 2:still acceptable, 3:satisfactory, 4:high), thus closely meeting the recommendations of this reviewer above. This status of regular physical activity was then used as predictor of CRF determined by the results of the cycle ergometer testing as outlined in subsection 2.2, which is thus a different variable to be clearly distinguished from the status of regular physical activity. In order to clarify this issue, we have inserted ‘regular’ before ‘physical activity’ at the adequate places in the methods, results and discussion sections, respectively.

Concerning the hint for the discussion, we have added the following text on the interplay between sedentary life style, physical inactivity, obesity and the immune system with respect to CRF to the corresponding paragraph in subsection 4.2 of the discussion:

“The age-related increase in the probability for low CRF, which was observed in univariate analyses, vanished when adjusting for the covariates, indicating that a decline of CRF with age observed at the population level might be linked to a mutual interplay between physical inactivity, often associated with a sedentary lifestyle, with obesity and the immune system [8,13-15,27].”

Please also refer to the attachment and the tracked changes detailed in the revised manuscript.

Round 2

Reviewer 1 Report

The authors have adequately addressed my comments

Reviewer 2 Report

I think the work is suitable for publication

Reviewer 3 Report

My all concerns were addressed.